# Verbal Fluency in Metabolic Syndrome

**DOI:** 10.3390/brainsci12020255

**Published:** 2022-02-12

**Authors:** Marcin Gierach, Anna Rasmus, Edyta Orłowska

**Affiliations:** 1Department of Endocrinology and Diabetology Collegium Medicum, Nicolaus Copernicus University, Skłodowskiej-Curie 9, 85-094 Bydgoszcz, Poland; 2Cardiometabolic Center Gierach-Med, 85-796 Bydgoszcz, Poland; 3Faculty of Psychology, Kazimierz Wielki University, 85-094 Bydgoszcz, Poland; ania.rasmus@gmail.com; 4Institute of Psychology, University of Gdańsk, 80-952 Gdańsk, Poland; edyta.orlowska@ug.edu.pl

**Keywords:** metabolic syndrome, insulin resistance, obesity, verbal fluency

## Abstract

Metabolic syndrome (MetS) or otherwise insulin resistance (IR) is described as a cluster of several commonly occurring disorders, including abdominal obesity; lipids disorders, such as hypertriglyceridemia; and low levels of high-density-lipoprotein cholesterol (HDL-C), hypertension (≥130/85 mmHg), and carbohydrates disorders, such as impaired fasting glucose or diabetes mellitus type 2. Type 2 diabetes (T2DM) constitutes insulin resistance, which is a strong risk factor for strokes. Patients with MetS are often prone to cognitive decline. Metabolic risk factors, hypertension, and diabetes, amongst them, have been hypothesized to play a great role in the pathogenesis of Alzheimer’s disease (AD) and the development of vascular dementia. For neuropsychological diagnostic and theoretical purposes verbal fluency is defined as a cognitive function that facilitates information retrieval from memory. It engages executive control and other cognitive processes, such as selective attention, selective inhibition, mental set shifting, internal response generation, and self-monitoring, as well as imagination and psychomotor skills. A total of 90 subjects, divided into 2 groups, patients with MetS (45) and healthy controls (45), were assessed. A significant difference in performance was found between the patients and controls, both in the phonetic (*p* < 0.01) and semantic fluency trials (*p* < 0.001). The MetS patients produced less words in the letter K and animal categories. The analysis of descriptive statistics shows that the group of patients with metabolic syndrome generated fewer words in both the phonetic and semantic categories. Our study shows that there is an association between metabolic factors and the verbal fluency performance of MetS patients. This is true, especially for phonetic verbal fluency, which is traditionally connected with the frontal cortex. Lower switching signifies possible executive dysfunctions amongst people with MetS. Subjects with this condition generated more diverse words and created less standard associations. This further implies the existence of dysexecutive syndrome and the need for diagnosing patients in this direction and involving this group of people in therapy. The proper correction of MetS components may improve cognitive function.

## 1. Introduction

Metabolic syndrome (MetS), or otherwise insulin resistance (IR), is described as a cluster of several commonly occurring disorders, including abdominal obesity; lipids disorders, such as hypertriglyceridemia; and a low level of high-density-lipoprotein cholesterol (HDL-C), hypertension (≥130/85 mmHg), and carbohydrates disorders, such as impaired fasting glucose or diabetes mellitus type 2 [1]. Type 2 diabetes (T2DM) constitutes insulin resistance, which is a strong risk factor for strokes [2]. Patients with MetS are often prone to cognitive decline. Researchers link MetS with a dysfunction in global cognition and increased dementia risk [3,4,5,6]. Nevertheless, there is no agreement on the specific domains that are impaired [7]. Many studies demonstrated a strong association between MetS and lower frontal lobe functions, such as executive functions attention and verbal fluency [7,8,9,10]. Some, however, failed to confirm this connection [11,12]. When it comes to other core cognitive domains, including episodic memory, perceptual speed, or visuospatial abilities, there is even less consensus, as available research studies are not consistent with each other [7,13]. Metabolic risk factors, for instance, hypertension and diabetes, have been hypothesized to play a great role in the pathogenesis of Alzheimer’s disease (AD) and the development of vascular dementia [1,14,15]. According to some researchers, the patomechanism of this cognitive decline can be the alteration of amyloid β-peptide (Aβ) metabolism, with increased amyloid deposition and increased phosphorylation of the tau protein. However, hypothetically hyperinsulinemia increases the risk of cognitive impairment through microvascular abnormalities. Brain subcortical small-vessel lesions are unlikely to cause cognitive impairment, but rather cause a syndrome characterized by neurologic and neuropsychological features, such as parkinsonism, gait disorders, relative sparing of memory, and dysexecutive syndrome (DEX) [16,17]. As far as neuropsychology is concerned, the last symptom is most promising and has become a subject of great interest. What is interesting, is that although most researchers apply different sets of methods, they usually include verbal fluency in them. Despite that fact, there are few data on metabolic syndrome and cognition [1], and even fewer focusing on verbal fluency itself, which is a complex topic.

For neuropsychological diagnostics and theoretical purposes, verbal fluency is defined as a cognitive function that facilitates information retrieval from memory. It engages executive control and other cognitive processes, such as selective attention, selective inhibition, mental set shifting, internal response generation, and self-monitoring, as well as imagination and psychomotor skills [18,19,20,21,22,23]. Tests of verbal fluency evaluate an individual’s ability to retrieve specific information within restricted search parameters, usually within a one minute time limit. Both classical and current studies show a multitude of various factors affecting the performance in verbal fluency tasks, and the complexity of psychological and neuronal mechanisms underlying it [24,25]. The aim of our study is to assess the impact of metabolic disorders on the verbal fluency in patients diagnosed with MetS.

## 2. Materials and Methods

A total of 90 subjects, divided into 2 groups, patients with MetS (45) and healthy controls (45), were assessed. All patients in the MetS group (18 females, 27 males) were recruited from the Department of Endocrinology and Diabetology in Bydgoszcz, Poland. The diagnosis of the MetS group was made on the basis of the International Diabetes Federation (IDF) criteria (Table 1).

The control group was recruited from visiting family members of other patients with light injuries treated in the same hospital. The controls were matched to the MetS patients according to gender, age, and education. Normal subjects were excluded from the study if their medical history included any major medical, neurological or psychiatric illness, substance abuse, or history of head injury or stroke.

All the subjects voluntarily provided written informed consent to participate in the testing sessions and were subjected to the same psychological examination in the form of the Mini Mental State Examination (MMSE) and verbal fluency tasks. METS patients scored lower (between 24 and 26 points) on the MMSE than the controls (full score), but still presented no symptoms of dementia. All patients included in the study showed no symptoms of increased anxiety or depression in the Hospital Anxiety and Depression Scale (HADS). Polish was the first language for all the individuals, and they were all right-handed.

Procedure: the authors used tasks from the Controlled Oral Word Association Test (COWAT), which is the most popular verbal fluency test and is one part of the Halstead–Reitan Neuropsychological Battery. It measures the spontaneous production of words belonging to the same category or beginning with some designated letter. The total number of words generated in 1 min for the letter K (phonemic fluency) and animal names was obtained from all 45 participants. The subjects were instructed that proper nouns were not acceptable. The instructions included reminding them not to use proper nouns. Each individual’s conduct was recorded both on paper and on tape, which limited the chance of examiners’ mistakes.

All procedures were performed after 12 h of fasting. Anthropometric measurements, including height, weight, and waist circumference (WC), were obtained for all the participants, as well as the blood pressure and body mass index (BMI) measured using the metric system. The BMI was calculated as the body weight (in kilograms) divided by the square of the body height (in meters). The WC was measured by placing a measuring tape around the waist at the upper point of the iliac crest. Finally, the following demographic factors were determined: age, sex, and obesity. The systolic and diastolic blood pressure were measured in the sitting position after 15 min of rest using an appropriately sized cuff in both upper extremities. Patients were seated quietly with their feet on the floor and the blood pressure readings were taken at 1 min intervals. An average of both measurements was calculated and used for the data analysis. Arterial hypertension was diagnosed according to the IDF definition (RR ≥ 130/85 mmHg). The levels of fasting total plasma cholesterol (TC), triglycerides (TG), high-density-lipoprotein cholesterol (HDL-C), and fasting blood glucose (FBG) were evaluated in all the patients. Low-density-lipoprotein cholesterol (LDL-C) was calculated using the Friedewald formula. Non-high-density-lipoprotein cholesterol (non-HDL-C) was figured on the base of the following formula: TC − HDL-C. In patients with abnormal fasting glycemia values and a waist circumference >80 cm in women or >94 cm in men, an oral glucose tolerance test (OGTT) was performed to determine glycemia in the fasting state and 2 h after the administration of 75 g of glucose.

Venous blood samples were collected from fasting patients for biochemical analyses (morphology, ionography, BUN, glucose, TSH, CRP, and fibrinogen). Abnormal results excluded the patient from the study.

Exclusion criteria: a history of heart surgery or other cardiovascular interventions, congenital defects of the heart, cardiac rhythm disorders, pregnancy, electrolyte disorders, inflammation, anemia, prostate disease, and Cushing’s syndrome. A self-reported history of medical and psychiatric problems, including a list of all currently prescribed medications, was obtained from each participant. Additionally, subjects after stroke or with dementia were excluded, as were those presenting other conditions that compromise cognition, such as depression, anxiety, taking psychotropic drugs, psychiatric diseases, and history of alcohol or chemical addiction or uncorrected visual or hearing disorders.

In order to diagnose a patient with MetS, at least three out of five components must be present. Amongst the subjects that constituted our study group, 31% (*n* = 14) were diagnosed with 3 components, 40% (*n* = 18) with 4, and 29% (*n* = 13) with all 5. All the patients were diagnosed with obesity, 86.67% with HT, 66.67% with high pre-meal blood glucose levels, 60% with TG, and 70% with HDL-C. The characteristics of the study group are presented in Table 2.

Statistical analysis was carried out using commercially available software (SPSS for Windows v 10).

## 3. Results

The analysis of the descriptive statistics shows that the group of patients with metabolic syndrome generate fewer words in both the phonetic and semantic categories (Table 3).

In order to identify the major differences between the groups’ performances in fluency tasks, we initially carried out a first exploratory analysis. We performed a Student’s *t*-test by entering the basic scores in the form of the number of produced words in each task. A significant difference in the performances was found between the patients and controls, both in the phonetic (*p* < 0.01) and semantic fluency trials (*p* < 0.001). The MetS patients produced less words in the letter K and animal categories. In addition to estimating the apparent differences in verbal fluency between those two samples, we were interested in comparing the profile of verbal fluency conduct. In order to achieve that, we entered the difference between the phonemic and semantic fluency calculated by subtracting the number of produced words in one trial from the number of words produced in the other. Next, we performed the Student’s *t*-test to compare the verbal fluency general profile. We did not observe any differences between the METS patients and controls, as both produced less words in the phonetic fluency task.

As we were interested in a detailed analysis of verbal fluency conduct, we entered the cluster numbers produced by the patients and controls in both fluency trials, and used the Student’s *t*-test procedure to compare the performances of both samples. The METS patients produced significantly less clusters in the phonemic fluency task (*p* < 0.1), but there was no such difference in the semantic fluency task. What is interesting though, is that the METS patients produced more phonetic clusters in the semantic animal category trial (*p* < 0.5) than the healthy subjects. We observed no significant difference between those two samples in the number of semantic clusters produced in the animal category task.

Further analysis involved the use of the Spearman’s correlation test to measure the strength and direction of the monotonic association between the verbal fluency results, in both the semantic and phonetic tasks, and each key medical factor contributing to the MetS condition (Table 4).

The gathered results indicate a significant association between fasting glucose levels and the patient’s performance in the phonetic verbal fluency task. Higher TGs are associated with a worse performance in phonetic fluency tasks. However, lower HDLs correlate with worse semantic and phonetic fluency tasks.

## 4. Discussion

The complexity of verbal fluency neuronal bases was confirmed by many studies on brain activity, for which extent and intensity are dependent on the fluency task type (phonemic, semantic, verbs, and association) [18,26], the addressed category characteristics (wide with many examples, or narrow with few of them), or the strategy chosen by the tested subject. As a result of the facts described above, it is not clear what parameters of verbal fluency realization should be considered as the most useful in clinical diagnoses. Without a doubt, most authors analyze the number of produced words. Nevertheless, in addition to that basic factor, one can also consider a number of mistakes (perseveration, words unrelated to the given criteria, and neologisms), a number of clusters (groups of words connected semantically or phonemically), and switching (between clusters). The last factors are proven to be connected with different cognitive resources, such as semantic memory, working memory (number of clusters), and executive functions (switching).

Furthermore, studies show the differences in verbal fluency realization amongst subjects with various brain dysfunctions [27,28]. In particular, numerous research analyzes verbal fluency deficits in dementia of various etiologies [29,30]. The left hemisphere dysfunction is usually associated with producing less words in phonemic than in semantic tasks, while the right hemisphere dysfunction is characterized by an opposite pattern, reaching beyond the presented criteria and producing main class names rather than examples [31]. This effect is explained by a thesis claiming that both brain hemispheres are responsible for different data processing strategies. Since the left brain hemisphere engages with typical, routinized, phonemic strategies, it is related to phonemic fluency and forming words into according clusters (consistent semantically or inconsistent with the task, such as phonemic clusters in semantic tasks). The right brain hemisphere is associated with heuristic, metaphoric, holistic, and perception-oriented strategies, which shape semantic fluency and also determine the choice of semantic strategy in phonemic tasks. Neuroimaging confirms that semantic fluency tasks (such as producing the names of animals) activate posterior brain arrears, mostly in the right hemisphere, primarily associated with using perception-oriented criteria. Brain area activation, however, is dependent on the given instructions, as producing tool names is mostly associated with the right parietal lobe. While both the left and right hemisphere strategies are usually effectively used by neurologically healthy individuals in every fluency task, the first one seems to be preferred in the beginning (up to 30 s) and the second follows it until the end [31].

There are data that point to the adverse effects of MetS on cognitive function [32]. Unfortunately, the mechanism by which MetS affects cognitive function, and which of its components play the main role, is still not well established. Some investigators indicate that high blood sugars are associated with the worsening of cognitive functions. A chronic high blood sugar level may decrease the synthesis of cerebral acetylcholine and cause the loss of neurons in the cerebral cortex [33,34]. In our study, the significant association between higher fasting glucose levels and the patient’s lower performance in phonetic verbal fluency tasks was shown. A similar observation was made by other researchers. Hye-Mi Oh et al. reported that among the metabolic risk factors, fasting plasma glucose affected the score of the neuropsychological assessment battery as CERAD-K (the Korean version of the Consortium to Establish a Registry for Alzheimer’s Disease used for the early diagnosis of dementia) [32]. Worse metabolic control in patients with diabetes had an impact on the cognitive functions. Avadhani et al. observed that higher HbA_1c_, which is a measure of average glucose concentrations over 2 months, is associated with a lower cognitive function in T2DM [35]. Limited studies have considered the relationship of TGs with cognition, and the findings are inconsistent. Among them, two studies concluded that the level of TGs was lower in patients with dementia or AD [29] compared with control groups, while several found no relationship between TGs and cognition [21,30]. Our results indicate an association between a worse performance in phonetic fluency tasks and higher levels of TGs. This observation is similar to the results that point to the inverse relation of TG levels and performances on various cognitive measures [31,32]. Hypertriglyceridemia may change cerebral blood by increasing the viscosity of blood and by causing arteriosclerosis. In our study, lower HDL was associated with worse semantic and phonetic fluency tasks. This finding correlates with the results of the Maine–Syracuse study, where higher HDL cholesterol was associated with better cognitive function. Significant positive associations were observed between HDL-cholesterol and the global composite score, working memory, and the MMSE after adjustment for demographic and cardiovascular risk factors. Participants with desirable levels of HDLs (≥60 mg/dL) had the highest scores for all cognitive outcomes [32]. Similarly, Crichton et al. observed that high HDL-C concentrations were associated with a better memory performance [36]. Some limitations have to be acknowledged. The education level in this group was rather low. Most subjects finished their education at secondary level or even before that. Higher education can negatively influence the association between MetS and cognitive decline [7]. Similar to most studies concerning MetS, we also focus on the older population. Studies involving younger subjects show mild cognitive changes associated with MetS or even none at all [37]. Age can be a factor indicating the progress dynamics of metabolic and cardiovascular conditions that impair cognition, starting from fluid functions traditionally associated with the frontal brain. While recognizing the limitations of this study, the results are significant to warrant additional research.

## 5. Conclusions

Our study shows that there is an association between metabolic factors and the verbal fluency performance of MetS patients. It is true, especially for phonetic verbal fluency, which is traditionally connected with the frontal cortex, that lower switching signifies possible executive dysfunctions amongst people with MetS. The subjects with this condition generated more diverse words and created less standard associations. This further implies the existence of dysexecutive syndrome and the need for diagnosing patients in this direction and involving this group of people in therapy. The proper correction of MetS components may improve cognitive function.

## Figures and Tables

**Table 1 brainsci-12-00255-t001:** IDF criteria of metabolic syndrome.

Abdominal obesity (cm)	F ≥ 80 or M ≥ 94
Arterial hypertension (HT) (mm Hg)	≥130/85 or treated for arterial hypertension
Triglycerides (TG) (mg/dL)	≥150 [1.7 mmol/L] or treated for dyslipidemia
HDL-C (mg/dL)	<50 [1.3 mmol/L] in women and <40 [1.0 mmol/L] in men
Fasting glycemia (mg/dL)	≥100 [5.6 mmol/L] or treated for diabetes

**Table 2 brainsci-12-00255-t002:** The characteristics of the study group.

Parameters	Total	Female	Male
N (%)	45	18 (40%)	27 (60%)
Age (y) ± SD	65.1 ± 4.8	63.8 ± 4.1	65.9 ± 5.3
BMI (kg/m^2^)	31.8 ± 4.8	31.2 ± 4.9	32.2 ± 4.7
WC (cm)	108.4 ± 14.1	106.2 ± 13.9	112.2 ± 15.4
SBP (mmHg)	146.7 ± 19.8	146.2 ± 19.2	147.1 ± 20.2
DBP (mmHg)	94.6 ± 6.6	94.1 ± 6.2	95.3 ± 7.2
TG (mg/dL)	149.2 ± 46.2	148.3 ± 45.7	150.4 ± 46.9
HDL-C (mg/dL)	41.7 ± 9.8	42.7 ± 9.3	41.1 ± 10.2
LDL-C (mg/dL)	118.5 ± 36.7	117.4 ± 35.4	119.8 ± 37.3
Non-HDL-C (mg/dL)	147.8 ± 42.6	146.2 ± 39.8	148.9 ± 44.4
IFG (*n*; %)	14/45 (31.1%)	6/18 (33.3%)	8/27 (29.6%)
T2DM (*n*; %)	16/45 (35.5%)	7/18 (38.8%)	9/27 (33.3%)
Education	8.7 ± 4.9	8.5 ± 4.8	8.8 ± 5.1
MMSE	25.43 ± 1.6	24.52 ± 1.4	25.92 ± 1.7

**Table 3 brainsci-12-00255-t003:** Difference in the results of verbal fluency amongst people with metabolic syndrome and controls.

Analysis	Average MetS Group	SD	Average Control Group	SD	*p* < 0.001
PHONETIC CATEGORY/1 min	11.73	3.81	15.86	2.27	**0.00004**
K—phonetic clusters	1.90	1.42	4.06	1.01	**0.00001**
K—phonetic switching	7.63	3.89	7.76	1.52	0.78424
K—semantic clusters	2.06	1.94	1.96	1.12	0.80854
K—semantic switching	8.68	3.87	12.13	3.14	**0.00027**
SEMANTIC CATEGORY/1 min	15.43	4.84	17.36	1.97	0.04765
Animals—phonetic clusters	1.36	1.03	0.66	0.84	0.01190
Animals—phonetic switching	12.5	5.05	15.1	2.12	0.01119
Animals—semantic clusters	3.93	1.41	4.46	1.3	0.13434
Animals—semantic switching	7.5	3.35	7.36	2.07	0.92779

**Table 4 brainsci-12-00255-t004:** Spearman’s correlation coefficient between the phonetic and semantic fluency tasks, and the medical parameters significant for metabolic syndrome diagnosis.

Parameters	Phonetic Task	Semantic Task
Waist circumference	0.15857	−0.09736
Systolic blood pressure	0.07991	−0.16656
Fasting plasma glucose	0.31329 *	0.15545
HDL-C	−0.32786 *	−0.30980 *
Triglycerides	−0.31573 *	−0.00444

* significant difference with *p* < 0.05.

## Data Availability

The study data may be available on request.

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
