# Peer review of "Verbal Fluency in Metabolic Syndrome"

_brainsci, 2022, doi:10.3390/brainsci12020255_

Round 1

Reviewer 1 Report

The study concerns patients with Metabolic syndrome (MetS), who are often prone to cognitive decline. For neuropsychological diagnostic and theoretical purposes verbal fluency is defined as a cognitive function that facilitates information retrieval from memory. It engages executive control and other cognitive processes. The study shows there is association between metabolic factors and verbal fluency performance of MetS patients.

The abstract covers the main aspect of the work. The introduction provides backround and information relevant to the study. The introduction maybe strengthen with more cause-effect relationship data. The methods are clear and replicable. The results presented match withe described methods. The results are novel and the study provides new knowledge. The data are plausible. The described findings by the author correlate with the results and the findings are relevant. The discussion (lines 209-238) maybe include more information and study data. The conclusions correlate to the results.

The figures and the tables are clear and legible. The paper does not raise any concerns. The manuscript does not raise any ethical concerns. The statistical analysis is appropriate to the research.

The references are relevant to the study and in the correct style. There are no concerns regarding similarities to other articles published by the same authors.

Author Response

We’d like to thank You for your valuable comments and suggestions. After a detailed analysis of comments, we have made the necessary changes to the manuscript. We also made editing of English language.

Yours faithfully,

Marcin Gierach

  1. The introduction provides backround and information relevant to the study. The introduction maybe strengthen with more cause-effect relationship data.

We added changes in manuscritp (line 42-47)

  1. The methods are clear and replicable. The results presented match withe described methods. The results are novel and the study provides new knowledge. The data are plausible. The described findings by the author correlate with the results and the findings are relevant. The discussion  maybe include more information and study data. The conclusions correlate to the results.

We added changes in manuscritp (line 203-209)

Reviewer 2 Report

This is a well written and well conducted study. I would recommend more information on metabolic syndrome and verbal fluency in the introduction. 

There are also a number of grammtical issues throughout the paper. A thorough proof read is recommended. 

Author Response

Dear Reviewer,

We’d like to thank You for your valuable comments and suggestions. After a detailed analysis of comments, we have made the necessary changes to the manuscript. We also made editing of English language.

Yours faithfully,

Marcin Gierach

REVIEWER 

  1. The introduction -more information on MetS and vertebral fluency

We added changes in manuscritp (line 42-47)

We added changes in manuscritp (line 203-209)

We made editing of English language.

Reviewer 3 Report

As a combination of metabolic syndrome (MetS) and insulin resistance, metabolic syndrome refers to a range of conditions including abdominal obesity, lipid disorders such as hypertriglyceridemia, hypertension, and carbohydrate disorders such as impaired fasting glucose or diabetes mellitus type 2. Diabetes type-2, results in insulin resistance, a common risk factor for stroke. Those who have MetS are more likely to have cognitive decline. In the pathogenesis of Alzheimer disease (AD) and the development of vascular dementia, metabolic risk factors, such as hypertension and diabetes, have been hypothesized to play a major role. As a cognitive function that facilitates the retrieval of information from memory, verbal fluency is defined for neuropsychological diagnostic and theoretical purposes.    As part of the study, authors assessed 90 subjects, divided into 2 groups -- 45 patients with MetS and 45 healthy controls -- to assess executive control and other cognitive processes. The two groups were divided into two subgroups -- those with MetS (45) and those without MetS (45).  The authors found significant differences in patient and control performance in both phonetic (p*0.01) and semantic fluency tests (p*0.00). Specifically, MetS patients produce fewer words in the letter K and animal categories. Furthermore, the authors showed there is an association between metabolic and verbal fluency performance among patients with metabolic syndrome based on descriptive statistics. This is especially true for phonetic verbal fluency, which is traditionally associated with the frontal cortex. Among those with MetS, lower switching indicates potential executive dysfunction. This condition was associated with more diverse words and fewer standard associations. Overall, I found the manuscript very appealing to read.    In addition, I encourage the manuscript to be published after addressing the following minor comments.  1. How these findings will be applied to diagnose young patients as the authors' participants are 65+/-5 years old. Would it be possible for authors to present data on a few young patients with disregard for any neurodegenerative disorders? 2. Authors should take care of language grammatical and typo errors throughout the manuscript. 

Author Response

Dear Reviewer,

We’d like to thank You for your valuable comments and suggestions. After a detailed analysis of comments, we have made the necessary changes to the manuscript. We also made editing of English language.

Yours faithfully,

Marcin Gierach

  1. How these findings will be applied to diagnose young patients as the authors' participants are 65+/-5 years old. Would it be possible for authors to present data on a few young patients with disregard for any neurodegenerative disorders?

We added changes in manuscritp (line 203-209)

  1. Authors should take care of language grammatical and typo errors throughout the manuscript. 

We made editing of English language.

Round 2

Reviewer 3 Report

Thanks to the authors to take care of my concerns.

I wish you good luck with your future research.